# Taguchi Grey Relational Analysis for Multi-Response Optimization of *Bacillus* Bacteria Flocculation Recovery from Fermented Broth by Chitosan to Enhance Biocontrol Efficiency

**DOI:** 10.3390/polym14163282

**Published:** 2022-08-12

**Authors:** Selena Dmitrović, Ivana Pajčin, Nataša Lukić, Vanja Vlajkov, Mila Grahovac, Jovana Grahovac, Aleksandar Jokić

**Affiliations:** 1Faculty of Technology Novi Sad, University of Novi Sad, Bulevar cara Lazara 1, 21000 Novi Sad, Serbia; 2Faculty of Agriculture, University of Novi Sad, Trg Dositeja Obradovića 8, 21000 Novi Sad, Serbia

**Keywords:** chitosan, flocculation, *Bacillus* sp., biocontrol, grey relational analysis, Taguchi method

## Abstract

Degradation of environment is a challenge to crop production around the world. Biological control of various plant diseases using antagonistic bacteria is an encouraging alternative to traditionally used chemical control strategies. Chitosan as a well-known natural flocculation agent also exhibits antimicrobial activity. The goal of this study was to investigate a dual nature of chitosan in flocculation of *Bacillus* sp. BioSol021 cultivation broth intended for biocontrol applications. Experiments were performed based on L_18_ standard Taguchi orthogonal array design with five input parameters (chitosan type and dosage, pH value, rapid and slow mixing rates). In this study, the grey relational analysis was used to perform multi-objective optimization of the chosen responses, i.e., flocculation efficiency and four inhibition zone diameters against the selected phytopathogens. The results have indicated a great potential of a highly efficient method for removal of the *Bacillus* bacteria from the cultivation broth using chitosan. The good flocculation efficiency and high precipitate antimicrobial activity against the selected phytopathogens were achieved. It has been shown that multiple flocculation performance parameters were improved, resulting in slightly improved response values.

## 1. Introduction

Plant diseases, caused by pathogenic bacteria and fungi, can lead to a reduction in plant growth or even death which leads to huge economic losses, as well as food safety issues [1]. The use of chemical fungicides and bactericides for management of plant pathogens results in environmental and health problems. That is the reason why the usage of biological control agents has been increasingly investigated [2,3,4]. Today, development of new methods to combat plant pathogens is imperative while the application of biopesticides could generate a critical change for the ecological and economical sustainability of the current food production and security [1].

According to a number of reports, bacteria from the *Bacillus* genus express several mechanisms making them suitable candidates to be applied as biological control agents of different plant pathogens [5,6]. Majority of bacteria of the genus *Bacillus* are Gram-positive, aerobic, endospore-forming and rod-shaped bacteria, which are found in diverse environments [5,6,7]. Among various bacterial antagonists reported, *Bacillus* spp. such as *B. amyloliquefaciens, B. cereus, B. licheniformis, B. subtilis* and *B. velezensis* have been used for effective control of the diseases by producing a variety of biologically active compounds such as lipopeptides, antibiotics, volatile organic compounds and enzymes with a broad spectrum of activities toward phytopathogens [8,9]. *Bacillus* cells are capable of forming dormant spores (endospores) that are resistant to extreme conditions and thus can be easily formulated and stored [10].

The raising interest of scientific public and agricultural sector for development of biological solutions for plant protection is a consequence of a potential threat arising from overuse of synthetic chemical pesticides, as well as the lack of efficient chemical bactericides [11]. On the other hand, many examples of highly efficient biocontrol agents could be found in the literature, whose commercialization and real-life application hasn’t happened, mostly due to high market price. The main setback to wider biopesticides application is their higher price compared to chemical alternatives, hence biocontrol agents’ production technology should be techno-economically viable, with positive effects to environment preservation, competitiveness increase in agricultural sector, and consequently to food safety. In addition to a high investment cost for the equipment necessary for biotechnological part of the biopesticide production process, downstream processing cost make up to 60% of the overall production cost, while the formulation step is usually a bottleneck for successful commercialization of biocontrol agents [12]. Therefore, considerable attention has been devoted to develop simple, efficient and economical methods for biocontrol agents harvesting or separation, and different methods, such as centrifugation, coagulation, filtration and flocculation, have been used [13].

Flocculation is a process in which particles are destabilized and induced to come together, make contact and subsequently form large agglomerates. Before flocculation, the colloidal dispersion must be destabilized and coagulation must occur, so that the particles cannot be redispersed before separation [14]. Flocculation is a widely used technique in many industries, including water/wastewater treatment, pulp and paper industry, production of pharmaceuticals, mineral processing, and many more branches of the food industry [15]. Flocculating agents can be classified into three groups: organic polymeric bioflocculants (poly-acrylamide derivatives and poly-acrylic acid), inorganic flocculants (aluminum-sulfate and polyaluminum chloride) and bioflocculants (microbial polysaccharides, glycoproteins, proteins, lipids, nucleic acids, nucleoproteins) [16,17]. Synthetic flocculants, either organic or inorganic, have been used in industrial fields due to low production costs and high efficiency; however, their use can cause adverse impacts on environment and health [18]. Natural flocculants are mainly based on chitosan, starch, cellulose, tannins and microbial raw materials. They can be divided into three categories based on overall electrical charge character: cationic flocculants, anionic flocculants and no-ionic flocculants, depending on the nature of functional groups constituting their chemical structure [19]. Chitosan attracts negatively charged bacterial cells because it is positively charged in acidic environments [20,21]. These ions may reduce bacterial electronegativity and promote sedimentation by adsorbing polymer chains [22]. Several factors may contribute to the antimicrobial activity of chitosan-three models have been suggested so far. The first model has considered the interaction between the positively charged chitosan and negatively charged microbial cells. The interaction is intermediated by the electrostatic forces between the protonated NH_3_^+^ groups and the negative residues [23], seemingly by competing with Ca^2+^ for electronegative sites on the membrane surface [24]. The second proposed mechanism is binding of chitosan with microbial DNA, which leads to the inhibition of the mRNA and protein synthesis via the penetration of chitosan into the nuclei of the microorganisms [25,26]. The third mechanism is the chelation of metals, suppression of spore elements and binding to essential nutrients for microbial growth [27]. Overall, such mechanism is more efficient at higher pH values, where positive ions are bounded to chitosan, since the amine groups are unprotonated and the electron pair on the amine nitrogen is available for donation to metal ions [28].

Chitin and chitosan have been investigated as biopolymers with antimicrobial activity against a wide range of target organisms such as algae, bacteria, yeasts and fungi [29]. Chitosan is a biopolymer obtained from chitin, found in the exoskeleton of crabs and insects and widely distributed in the nature. In recent years, much attention has been paid to chitosan. The reasons for that are probably its biocompatibility, biodegradability, non-toxicity and some specific properties such as the role in biofilm formation, chelation and adsorption, as well as antimicrobial activity [30]. There are many factors that affect the antimicrobial activity of chitosan, including: type of test microorganism, the molecular weight of chitosan (Mw), the degree of deacetylation (DD) of the biopolymer chain, chitosan concentration, pH value, chitosan source, temperature, cell growth phase etc. [31]. Microorganisms could be divided into three groups: Gram-positive bacteria, Gram-negative bacteria and fungi [32]. According to some of the research results, chitosan has shown a more pronounced bactericidal effect against the Gram-positive bacteria (e.g., *Listeria monocytogenes, B. megaterium, B. cereus, Staphylococcus aureus, Lactobacillus plantarum, L. brevis, L. bulgaris,* etc.) than against the Gram-negative bacteria (e.g., *Escherichia coli, Pseudomonas fluorescens, Salmonella typhymurium, Vibrio parahaemolyticus*, etc.) [33], while some other research results suggest otherwise [31]. The opinions also differ as to whether chitosan had a stronger antimicrobial effect against bacteria or fungi [31,33,34,35,36]. Studies carried out using *B. cereus*, *E. coli*, *S. aureus, B. subtilis* and *L. monocytogenes* have proved that for the lower chitosan molecular weight (Mw) the greater was the observed effect on the reduction of the test microorganism’s growth and multiplication [29]. The higher deacetylation degree (DD) of chitosan leads to its higher antibacterial activity [37]. The antibacterial activity of chitosan largely depends on the pH value, not only because chitosan is soluble only in an acidic medium, but also because chitosan molecules become polycationic at pH < 6.5 [38]. The antibacterial activity decreases with the increase of pH value and it is completely lost at pH 7.0, probably due to the fact that amino groups of chitosan are no longer charged, and chitosan is poorly soluble in water at pH 7.0 [25,39]. Recently, some studies have shown that chitosan can be produced from fungi [40]. Temperature has a significant effect on the antibacterial activity of chitosan, as well as on its viscosity and Mw [39]. Chitosan shows enhanced bactericidal activity with increasing temperature ranging from 4 °C to 37 °C [23]. In general, the chitosan solution before storage has higher antibacterial activity than the chitosan solution stored for 15 weeks [41]. Similar to bacteria, the chitosan activity against fungi is assumed to be fungistatic, and generally chitosan has been reported as very effective in inhibiting spore germination, germ tube elongation and radial growth inhibition [42,43]. The antifungal mechanism of chitosan involves interference with cell wall morphogenesis, resulting in chitosan molecules direct interference with fungal growth [42]. Chitosan oligomers diffuse inside hyphae interfering the enzymes activity responsible for fungal growth [44]. The intensity of chitosan activity when it comes to degradation of fungal cell walls depends on its concentration, DD and pH value. Studies conducted using cultures of *Rhizoctonia solani* and *Sclerotium rolfsii* on nutrient agar revealed that the percentage of fungal growth decreased with increase of the chitosan concentration in the medium. As the inhibition process takes place, the medium shifted towards alkalinity which decreases the chitosan effectiveness [45]. In addition to its antimicrobial activity, chitosan application in agriculture is based on its additional beneficial traits including cationic nature, biodegradability, non-toxicity and adsorption properties [46]. Chitosan could be used as a carrier for active components of biopesticides and (bio)fertilizers, also contributing to prevention of fertilizer losses [47]. It could be as well applied as seed coating agent, plant immunity elicitor and soil amendment [48], resulting in improved plant physiological state and longer post-harvest shelf life of agricultural products. Chitosan effect on plant immunity and resistance induction against the pathogens is of utmost importance, considering that chitosan contributes to activation of plant defense signaling pathways [47] inducing both local and systemic plant defense responses. Furthermore, chitosan provides better water availability for plants, as well as increased plant tolerance to drought, salinity and high temperature [49], thus contributing to management of plant abiotic stress [46]. All of the aforementioned reasons support application of chitosan as both flocculation agent, used for downstream separation of bacterial antagonists as potential biocontrol agents, and a formulation agent with several beneficial traits for the final biocontrol product intended for application in agricultural practice. In most cases, chitosan-based formulations of bioproducts for agriculture could be applied as solid formulations (soil amendment) or after resuspending in water (seed coating, foliar spraying and post-harvest fruit and vegetable coating agents) [50].

Most of the studies mentioned above have been focusing on predicting the optimal processing parameters aimed at chitosan flocculation efficiency or its antimicrobial activity. Previous research has confirmed antifungal activity of the producing microorganism *Bacillus* sp. strain BioSol021 against phytopathogenic and mycotoxigenic fungi *Aspergillus flavus* [51]. Toxigenic *Aspergillus* species are contaminants of agricultural commodities including three leading agricultural crops: wheat, rice, and maize. Therefore, these species pose a burden to food and feed safety, international trade and human and animal health [52]. However, wide spectrum of antimicrobial activity is a highly desirable trait of biocontrol agents or biopesticides in agricultural practice. Hence this study has investigated a possible spectrum of antimicrobial activity of the biocontrol agent based on *Bacillus* sp. strain BioSol021 using test microorganisms from the genera *Fusarium*, *Colletotrichum,* and *Xanthomonas* as representative fungal and bacterial phytopathogens, besides the fungi *Aspergillus flavus*. The genus *Fusarium* comprises of fungal species causing different types of plant diseases, including head blight, vascular wilt, canker, root rot, infecting over 100 economically important crops and presenting a threaten to food chain safety considering their ability to produce mycotoxins [53]. Fungal pathogens from the genus *Colletotrichum* represent a major threat to fruit, cereals, leguminous and other crops and this genus is ranked among the top 10 fungal pathogens according to State of the World’s Plants 2016 [54]. *Xanthomonas* is a well-known genus of bacterial plant pathogens infecting mostly economically important vegetable crops [55]. Plant disease severity followed by crop losses could be up to 100%, with additional problem of fast resistance development by bacterial phytopathogens to most of the commercially applied bactericides [56,57]. Suppression and management of bacterial plant diseases is a major challenge in agricultural practice. The lack of efficiency when it comes to copper-based bactericides, the forbidden usage of antibiotics in agriculture in majority of the countries worldwide and the rising emergence of pesticide-resistant bacterial phytopathogenic strains have resulted in necessity to develop safe and highly efficient bactericides which could be successfully commercialized and available to agricultural community [58]. One of the examples pointing out the necessity to develop novel bactericides is the fact that there are no products in the category of biobactericides registered and allowed to use in organic agriculture in the Republic of Serbia [59]. The published data regarding the bioprocess development when it comes to biopesticides production are mostly lab-scale data with low application potential and techno-economic viability. Therefore, the aim of this study was to investigate the novel downstream procedure for biocontrol agents’ production based on dual chitosan influence on both *Bacillus* sp. BioSol021 broth flocculation, as well as the antimicrobial activity of the bacteria/chitosan mixtures, i.e., precipitate, against the wider spectra of fungal and bacterial plant pathogens. The effects of the chitosan type and dosage, pH, rapid and slow mixing rates during flocculation were investigated to define the optimal conditions for the flocculation procedure. L_18_ Taguchi orthogonal array was selected as a practical way for outlining the best combination of different variables in such phenomenon requiring an excessive number of experiments. A multi-response grey relational analysis (GRA) was employed to identify parameters that simultaneously affect reaching the optimized efficiency of the flocculation and the antimicrobial activity of the final flocculation product.

## 2. Materials and Methods

### 2.1. Microorganisms

Production microorganism used in this study was *Bacillus* sp. strain BioSol021, isolated from the rhizosphere of common bean or kidney bean (*Phaseolus vulgaris*) and identified using 16S rRNA gene sequencing (GenBank sequence accession number ON569805). Genomic DNA was isolated using a commercial kit (PureLink Genomic DNA kit, Thermo Fisher Scientific, Waltham, MA, USA). Isolated genomic DNA was used as a template to amplify the sequence of 16S rRNA gene using the universal pair of primers 27f (5′-AGAGTTTGATCCTGGCTCAG-3′) and 1492r (5′-GGTTACCTTGTTACGACTT-3′). The PCR (polymerase chain reaction) method was performed using PCR device (Surecycler 8800 Thermocycler, Agilent Technologies, Santa Clara, CA, USA) and the following composition of PCR mixture with a total volume of 50 μL: 2 μL genomic DNA, 1 μL forward primer, 1 μL reverse primer, 25 μL Master mix (AmpliTaq Gold™ 360 Master Mix—Thermo Fisher Scientific, Waltham, MA, USA), 21 μL of water (UltraPure Ultra DNase/RNase-Free Distilled Water, Invitrogen, Waltham, MA, USA). Amplified DNA fragments were sequenced in both directions (Macrogen Europe, Amsterdam, The Netherlands) using the same primers used in the amplification phase. The chromatograms obtained were visualized using FinchTV 1.4.0 software (Geospiza, Inc., Seattle, Washington, DC, USA). Sequences (forward and reverse) were then processed using the MEGA 7 software package (Molecular Evolutionary Genetics Analysis, Penn State University, State College, PA, USA) [60], and a consensus sequence with a total length of 1416 bp was obtained. Using the BLAST program (Basic Local Alignment Search Tool, BLAST; http://blast.ncbi.nlm.nih.gov/Blast.cgi (accessed on 5 March 2022)) the obtained sequence was compared with the deposited sequences available in the database (http://www.ncbi.nlm.nih.gov/genbank (accessed on 5 March 2022)). Phylogenetic analysis was performed using the obtained sequence of the production microorganism and 16S rRNA region of related *Bacillus* strains using MEGA 7 software [61]. During the construction of the phylogenetic tree, the maximum likelihood method with bootstrap analysis in 1000 replicates was used. In addition, the 16s rRNA gene sequence of *Streptomyces albidoflavus* as an outgroup microorganism was used to construct the phylogenetic tree. Biochemical analysis of the producing microorganism’s metabolic activity was performed using the VITEK 2 compact system and VITEK BCL test kit (bioMérieux, Marcy-l’Étoile, France).

Test microorganisms or phytopathogens were bacteria *Xanthomonas* sp., isolated from diseased pepper plants with symptoms of bacterial spot, fungi *Colletotrichum* sp. and *Fusarium* sp., isolated from apples with rot symptoms, and *Aspergillus* sp. with proven aflatoxigenic potential isolated from maize [52]. All pathogens were identified according to pathogenic, morphological and ecological characteristics. The phytopathogenic *Xanthomonas* isolate identification was confirmed by the PCR (polymerase chain reaction) method using the species-specific primers (XeF and XeR) [62]. Appearance of 208 bp-fragments during observation by UV transluminator was considered as a positive reaction for identification of the pathogen as member of the *X. euvesicatoria* species [63]. The identification of *Colletotrichum* sp. was confirmed by the PCR method, using species-specific primers for *C. acutatum* CaInt2, in combination with the conserved primer ITS4, according to the protocol described by Screenivasaprasad et al. [64]. The occurrence of amplicon about 490 bp in size was considered a positive reaction for *C. acutatum* identification [65]. The identification of *Fusarium* sp. was confirmed by the PCR method using the species-specific primer pair FA-ITSF and FA-ITSR [66,67] and the amplified fragments were of the expected size (272 bp) for *F. avenaceum* [68]. *Aspergillus* section *Flavi* strain was identified to the species level based on macroscopic and microscopic morphological characteristics and marked as *A. flavus* strain SA2BSS [52]. All microorganisms used in this study are preserved in the Laboratory for Biochemical Engineering of the Faculty of Technology Novi Sad, University of Novi Sad.

Fungal test microorganisms (*Aspergillus* sp., *Colletotrichum* sp., *Fusarium* sp.) were kept on a semi-solid semisynthetic nutrient medium SMA (Sabouraud maltose agar), obtained by combining SMB (Sabouraud maltose broth, HiMedia Laboratories, Thane West, Maharashtra, India) and 2% (*w/v*) agar (HiMedia Laboratories, Thane West, Maharashtra, India); *Xanthomonas* sp. was kept on a semi-solid semisynthetic medium YMA (yeast maltose agar), containing (g/L) glucose (15), peptone (3), yeast extract (3), malt extract (1) and agar (20). All microorganisms were stored at 4 °C. In order to stimulate their metabolic activity and ability to reproduce, test microorganisms were refreshed by applying the same nutrient media that were used for their storage under the following conditions: *Xanthomonas* sp. at 26 °C for 48 h, while phytopathogenic fungi were refreshed at 26 °C for 120 h.

### 2.2. Inoculum Preparation and Cultivation Parameters

Production microorganism (*Bacillus* sp. strain BioSol021) was kept on a semi-solid commercial medium nutrient agar (HiMedia Laboratories, Thane West, Maharashtra, India). Preparation of the inoculum of *Bacillus* sp. strain BioSol021 was performed by seeding the liquid commercial medium, nutrient broth (HiMedia Laboratories, Thane West, Maharashtra, India), with previously refreshed microorganisms (28 °C for 48 h on nutrient agar) and by cultivation for 48 h at 28 °C, with stirring on a laboratory shaker (150 rpm) and spontaneous aeration. Cultivation of *Bacillus* sp. strain BioSol021 lasted 96 h at 28 °C in a 16 L-laboratory bioreactor (EDF – 15.4_1, A/S Biotehniskais centre, Riga, Latvia), with stirring rate of 100 rpm using the three impeller-Rushton turbine and aeration rate of 1.5 vvm (volume of sterile air/(volume of the medium·min)) [69]. Cultivation medium contained (g/L) cellulose (5.0), (NH_4_)_2_SO_4_ (3.8), K_2_HPO_4_ (0.3) and MgSO_4_·7H_2_O (0.3). Initial pH value of the cultivation medium was set to 6.0 prior to sterilization by autoclaving. Cell number at the end of cultivation was determined by counting the colony-forming units [70]. The carbon and nitrogen source content at the end of the cultivation were determined using the methods described by [71] and [72]. The produced cultivation broth of the producing strain *Bacillus* sp. BioSol021 was further used for investigation of different flocculation efficiency parameters.

### 2.3. Antimicrobial Activity

After the flocculation tests, antimicrobial activity of the precipitate against the selected phytopathogens was examined in vitro using the well diffusion method (fungal test microorganisms) and the diffusion-disc method (bacterial test microorganism).

#### 2.3.1. Well-Diffusion Assay

Antifungal activity of *Bacillus* sp. BioSol021-chitosan precipitate was tested using the well diffusion technique. In 90 mm Petri plates, a layer of mixture of the SMA medium (15 mL) and fungal pathogen suspension (1 mL–10^5^ CFU/mL), homogenized using the vortex mixer, was spread. After the medium solidification, three wells per plate with a diameter of 10 mm were made. In each well 100 μL of the flocculated sample was added. The assessment of antagonistic activity was carried out after 96 h of incubation at 26 °C by measuring the inhibition zone diameters (mm) [68].

#### 2.3.2. Disk-Diffusion Assay

Antibacterial activity of *Bacillus* sp. BioSol021-chitosan precipitate was tested using the diffusion-disc method. In 90 mm Petri plates, a layer of mixture of the YMA medium (15 mL) and bacterial pathogen suspension (1 mL–10^8^ CFU/mL), homogenized using the vortex mixer, was spread. After solidification, three disks per plate with a diameter of 6 mm were applied firmly on the medium surface. On each disk, 10 μL of the flocculated sample was added. The assessment of antagonistic activity was performed after 48 h of incubation at 28 °C by measuring the inhibition zone diameters (mm) [72].

### 2.4. Flocculation Experiments

The jar tests were applied to determine the optimal experimental conditions in terms of the selected flocculation process variables. Table 1 summarizes the experimental factors and their levels.

Different types of chitosan were provided by Sigma-Aldrich (St. Louis, MO, USA) (ch 1: obtained from shrimp shells, ≥75% deacetylated; molecular mass 100,000–300,000 Da; ch 2: ≥75% deacetylated, molecular mass 310,000–375,000 Da) and Acros-Organic (Geel, Belgium) (ch 3: ≥90% deacetylated, molecular mass 600,000–800,000 Da). Chitosan solutions were prepared by mixing 100 mg (dry weight) of different chitosan types with 100 mL of 0.1 M HCl solution with continuous stirring by the magnetic stirrer (25 °C, 100 rpm). Stirring was carried out until chitosan was completely dissolved in the solution. Only freshly prepared solutions (stored at 4 °C for less than 24 h) were used in all flocculation experiments.

The experiments were performed using 250 mL of the *Bacillus* sp. BioSol021 cultivation broth, obtained after the 96 h-bioreactor cultivation, in a 400 mL-beaker, in which the appropriate amount of the flocculant was added, according to the experimental plan given in Table 1. pH value was adjusted using 0.1 M HCl/0.1 M NaOH. The sample beaker was placed on a magnetic stirrer, first for 5 min of rapid mixing to disperse the flocculant and then the speed was reduced to slow mixing speed for 30 min to enhance particle collisions. The settling time was 30 min. All flocculation experiments were conducted at the same temperature (25 °C).

The flocculation efficiency was calculated by the spectrophotometric measurements of the microbial suspension optical density (absorbance at 600 nm, UV 1800, Shimadzu, Kyoto, Japan) prior and subsequent to flocculation experiments:(1)$$\%\mathrm{EF}=\frac{{\mathrm{OD}}_{0}-\mathrm{OD}}{{\mathrm{OD}}_{0}}\times 100$$
where OD_0_ and OD are the optical densities of the samples taken before and after the flocculation, respectively.

### 2.5. Taguchi Method and Grey Relational Analysis (GRA)

Taguchi method represents a practical way for outlining the optimal combination of different factors, especially in a phenomenon such as flocculation that requires substantial number of experiments to investigate the factors’ influences on the selected response. This approach helps to identify the influence of individual factors and establish valid relationship between the responses and the operational conditions [73,74]. The experimental design for five controllable variables (chitosan type, pH value, chitosan dosage, rapid and slow mixing rates) with three levels is organized by the Taguchi’s L_18_ orthogonal array. In Taguchi orthogonal L_18_ design, only 18 runs were needed, substantially reducing the number of runs over a full factorial experimental design [75]. The Minitab software package (version 17.0, Minitab, State College, PA, USA) was used for statistical analyses. In this study, further processing procedure should find optimal controllable variables for multi objective function using the grey relational analysis (GRA). In this way, the complex multi-objective optimization is simplified into a relational grade as a single response. For this purpose, five responses were chosen, i.e., flocculation efficiency and four inhibition zone diameters as indicators of the antifungal and antibacterial activity against the selected phytopathogens.

## 3. Results and Discussion

### 3.1. Identification of The Producing Microorganism

Identification of the producing microorganism was performed using 16S rRNA gene sequencing and VITEK2 identification. In order to define the genetic similarity of the tested isolate with the reference strains available in the database, phylogenetic analysis was performed by generating a phylogenetic tree (Figure 1). The maximum likelihood method was used for the construction of the phylogenetic tree, as a common statistical approach for estimating unknown parameters of the probability model based on the Tamura-Nei model [76].

The phylogenetic tree given in Figure 1 and the results of VITEK2 biochemical tests given in Table 2 have confirmed that the producing microorganisms *Bacillus* sp. strain BioSol021 belongs to the operative group *B. amyloliquefaciens* (BLAST similarity 100%, VITEK2 confidence 96–99%—excellent identification).

### 3.2. Taguchi Optimization of The Flocculation Process

Since *Bacillus* sp. BioSol021 cultivation broth produced in the 16 L-laboratory bioreactor was used as a biocontrol agent and initial mixture for the flocculation procedure, the composition of the cultivation broth was investigated at the end of the cultivation process to determine the main components and biomass content of the bacterial antagonist. The results are given in Table 3. As it could be seen, the cultivation broth contained 3.4 g/L of cellulose, while the total nitrogen content was 0.37 g/L. The biomass content of *Bacillus* sp. BioSol021 at the end of the cultivation and at the beginning of the flocculation was 2∙10^8^ CFU/mL, while the pH value was 7.51.

The actual flocculation efficiencies and the corresponding signal–to–noise (S/N) conversion of each run were determined for the proposed 18 experiments by the Taguchi method. Run 1 (boldfaced) had the maximum S/N ratio over the other experimental tests (Table 4). The parameter combination for this run was A_1_B_1_C_1_D_1_E_1_ (chitosan ch1, pH 5, flocculant dose 45 mg/L, rapid mixing 250 rpm and slow mixing 50 rpm).

Table 5 shows the ANOVA (analysis of variance) results of the S/N ratios for the flocculation efficiency, i.e., the data presented in Table 4. The obtained *p*-values for the selected flocculation factors considering the S/N ratios have indicated that the pH value has had the greatest effect on the flocculation efficiency, followed by the chitosan type. Both of the factors were statistically significant at the significance level of 95%.

Figure 2 illustrates the main effects plot for the S/N of the flocculation efficiency. Solid lines connect each mean value within each selected factor, while the dashed reference line specifies the overall mean. The main effect for the single variable is estimated by relating the solid plotted line to the dashed reference line. As it can be seen, pH value has shown the most significant influence on the flocculation efficiency, as it has shown steeper slope compared to the other factors.

The best flocculation efficiency was achieved at pH value 5. Chitosan is a weak polyelectrolyte, so its charge density is consequently significantly influenced by pH value [77]. The flocculation efficiency was lower at higher pH value. This observation is in a good agreement with the results found in literature [78,79]. Hughes et al. [78] have reported that lowering the pH value below 7.0 enables the flocculation of *B. subtilis* using chitosan. High pH value dependence of *B. amyloliquefaciens* flocculation efficiency by chitosan in the complex medium such as cultivation broth suggests a mechanism dependent upon the involvement of electrostatic interactions [78].

The flocculation processes is usually driven by two or more flocculation mechanisms [80]. Chitosan is a distinctive cationic polyelectrolyte with amino groups that are protonated in acidic aqueous medium [80]. Chitosan cationic behavior and Mw can be useful both for charge neutralization and for particle entrapment [81]. Due to the various structural characteristics and abundant functional groups of bacterial cells, the link between the chitosan structural factors (DD and Mw) and the flocculation efficiency of bacterial suspensions is an irregular one [77]. In this study, in the case of chitosan with the same DD, lower Mw chitosan, ch1 (compared to higher Mw chitosan ch2), had higher flocculation performance. Similar results were published for flocculation of anaerobic sludge [82]. Even though ch3 had higher Mw, flocculation efficiencies obtained for chitosan type 1 (ch1) and 3 (ch3) are similar probably due to higher DD (>90%) of ch3.

The flocculation efficiency exhibited a small increase when increasing the flocculant dosage. Cultivation broth is a complex matrix that contains various extracellular components, as well as residual medium nutrients, besides microbial cells. Chitosan interacts with all of these components and for all conducted experimental runs the flocculation efficiency was above 70%.

The main effects plot (Figure 2) shows speed of rapid mixing to be next to the overall mean (dash line), indicating that its impact on the flocculation efficiency is not significant in the selected experimental factor levels’ range. These results are consistent with the ANOVA analysis, Table 6. These results suggest that all the selected values of rapid agitation rate were sufficient to ensure equal dispersion of the chitosan into the cultivation broth to incite efficient flocculation [83,84]. On the other hand, the influence of slow mixing speed is higher compared to the rapid mixing speed, as it has steeper slope (Figure 2). Intensification of the slow mixing (higher speed) resulted in the flocculation efficiency decrease. The reason for this can be breaking up of the formed flocs through intense mixing.

From the data presented in Figure 2, the optimal flocculation parameters were A_3_B_1_C_3_D_3_E_1_, i.e., chitosan type 3 (ch 3), pH value 5, flocculant dose 65 mg/L, and rapid and slow mixing rate 450 rpm and 50 rpm, respectively.

### 3.3. Taguchi Grey Relational Analysis (GRA) for Multi-Response Optimization

As mentioned previously, chitosan is well known by its flocculation properties, but at the same time it has been used in biocontrol of various microbial pathogens in agriculture [84]. For this reason, antimicrobial activity of the precipitate, consisting of chitosan and flocculated *Bacillus* sp. BioSol021 cells, was examined against different bacterial and fungal phytopathogens (*Xanthomonas* sp., *Colletotrichum* sp., *Fusarium* sp. and *Aspegillus* sp.). The goal of this study was to achieve both high flocculation efficiency and high biocontrol activity of the flocculated cultivation broth after sedimentation. A multi-response optimization using Taguchi GRA was carried out to achieve this goal. The results of the precipitate antimicrobial activity tests alongside the flocculation efficiency are given in Table 6.

Data pre-processing is the first step in GRA, implemented to transfer the original sequence to a comparable sequence. Numerical data are normalized to the values between 0 and 1. The normalized value of the original sequence for all responses is larger-the-better performance characteristic and it can be expressed by the following equation:(2)$$X_{\mathrm{ij}}=\frac{Y_{\mathrm{ij}}{-\;\min(Y}_{\mathrm{ij}})}{\max\left(Y_{\mathrm{ij}}\right){-\;\min(Y}_{\mathrm{ij}})}$$
where X_ij_ is the normalized response value and Y_ij_ is the response value obtained from the experimental runs. There were five responses, so j had values j = 1, 2, 3, 4 and 5. The number of experimental runs i is the i-th experimental data (i = 1, 2, 3, …, 18). Values min (Y_ij_) and max (Y_ij_) represent the minimum and the maximum value of Y_ij_ for each response.

The next step in the GRA is calculation of grey relational coefficient (GRC) of each response using the following equation:(3)$${\mathrm{GRC}}_{\mathrm{ij}}=\frac{\left(\Delta_{\min}{+\psi \Delta }_{\max}\right)}{{(\Delta }_{\mathrm{ij}}{+\psi \Delta }_{\max})}$$
where GRC_ij_ is the grey relational coefficient for j-th response in the i-th experimental run. Deviation sequences, Δ_ij_, is treated as a quality loss from the target value, calculated as the difference between 1 and normalized value for each response. Distinguishing coefficient, ψ, takes values in range 0 to 1, and value of 0.5 was selected in this study.

Normalized values of the experimental data obtained after the normalization of the original data for the selected responses and their corresponding deviations are shown in Table 7.

The last step was to perform the grey relational calculation on multi-responses using Equation (4):(4)$${\mathrm{GRG}}_{i}=\frac{1}{n}\overset{n}{\underset{j=1}{\sum }}{\mathrm{GRC}}_{\mathrm{ij}}$$
where GRG_i_ is the grey relational grade (GRG) for the i-th experimental run and n is the number of responses, in this study 5.

In this paper, the goal of achieving a good flocculation efficiency and high precipitate antimicrobial activity against the selected phytopathogens were determined to be equally important, i.e., the same weight was given to all responses for the GRG calculation. The GRCs, GRG, and S/N conversion of GRG for each run are shown in Table 8, with higher values being more desirable. The 7th experimental setting (A_3_B_1_C_2_D_1_E_3_) was determined to be the preferred processing parameters combination (chitosan ch3, pH 5, flocculant dose 55 mg/L, rapid mixing 450 rpm and slow mixing 50 rpm), with a GRG value of 0.8730.

Table 9 shows the ANOVA analysis results of the S/N ratios for GRG. The obtained selected flocculation factors’ *p*-values for S/N ratios indicated that pH value had the greatest effect on GRG (statistically significant at the significance level of 95%), followed by chitosan type, slow mixing rate and dosage. Rapid mixing rate was the least significant factor affecting GRG.

Figure 3 illustrates the main effects plot for the S/N of the GRG. In contrast to the flocculation optimization, differences between ch1 and ch3 were more pronounced when inhibition zone diameters against the selected bacterial and fungal phytopathogens were taken into account as the additional responses. The antimicrobial activity of chitosan is greater for higher DD [37], in this study ch3. Another reason for the improved effect of this chitosan type, in the case of multi-objective optimization, can be found in Mw difference between the applied chitosan types. In previous studies it was found that lower chitosan Mw has adverse effect on microorganism growth and multiplication [29]. Hence in this case, chitosan with higher Mw could have beneficial effect on the precipitate antimicrobial activity against the selected bacterial and fungal plant pathogens.

As it can be seen from Figure 2 and Figure 3, pH value has identical trend for both optimization cases. The highest GRG values, as well as chitosan flocculation efficiency, were obtained for acidic environment (pH 5). Similar as the flocculation efficiency, chitosan antimicrobial activity is higher in acidic environment [25,39].

In the case of multi-response optimization, the optimal chitosan concentration was found to be 55 mg/L. These results are rather enthusiastic due to the lower chitosan consumption during the flocculation procedure, affecting the economic viability of the proposed downstream procedure.

The influence of slow mixing rate is higher in the case of multi-response optimization compared to its influence in the case of flocculation efficiency optimization alone. In the same instance, slow mixing rate has steeper slope (Figure 3) compared to the influence on the flocculation efficiency. The same values of mixing rates can be applied in the case of multi-response optimization.

From the data given in Figure 3, the optimal flocculation parameters in order to obtain maximized values of the flocculation efficiency and antimicrobial activity against the selected phytopathogens were A_3_B_1_C_2_D_3_E_1_, i.e., chitosan type 3 (ch 3), pH 5, flocculant dose 55 mg/L, and rapid and slow mixing rate 450 rpm and 50 rpm, respectively.

The A_3_B_1_C_2_D_3_E_1_ had been the optimal factors combination of the multi-response optimization obtained by the GRA and this combination of operational parameters was applied in the validation experiment. The results of the validation test are presented in Table 10.

A good agreement between the actual and the predicted GRG was obtained. It is shown that multiple performance aspects of the flocculation process were improved, resulting in slightly improved response values.

Literature data suggests there is a synergetic effect between chitosan and endophytic *Bacillus* species [84] when it comes to antimicrobial activity against plant pathogens. Chitosan antimicrobial activity was confirmed against selected phytopathogens using the well/disc diffusion assay for chitosan type 3 (ch 3) and concentration 55 mg/L. For all tested phytopathogens inhibition zone diameter was in range 13.5–15 mm. Further analyses were conducted to evaluate and quantify the possible antimicrobial activity synergy between the *Bacillus* sp. BioSol021 and chitosan. *Bacillus* sp. BioSol021 cultivation broth was centrifuged at 12,000 rpm for 10 min and precipitated biomass was resuspended to the final volume of flocculated samples (flocculation precipitate) for comparison of antimicrobial activity between the antagonist biomass itself and flocculated samples. The results showed that antimicrobial activity testing of the suspended *Bacillus* sp. BioSol021 biomass without chitosan addition has resulted in the following inhibition zone diameters against *Xanthomonas* sp., *Colletotrichum* sp., *Fusarium* sp. and *Aspergillus* sp. - 25.00 mm, 26.33 mm, 28.33 mm, and 30.33 mm, respectively. On the other hand, the results of antimicrobial activity testing of the flocculated samples showed the following values of inhibition zone diameters against *Xanthomonas* sp., *Colletotrichum* sp., *Fusarium* sp. and *Aspergillus* sp.—43.00 mm, 40.33 mm, 37.00 mm, and 44.85 mm, respectively. Higher antimicrobial activity of the flocculated samples suggested synergistic effect between the *Bacillus* sp. BioSol021 and chitosan in terms of the antagonistic effect against the tested plant pathogens.

## 4. Conclusions

In this study chitosan has proven to be an effective natural flocculant for removal of the *Bacillus* bacteria from the cultivation broth, reaching flocculation effectiveness up to 98.60%. Additionally, high precipitate antimicrobial activity against the selected phytopathogens was achieved. Experiments based on L_18_ standard Taguchi orthogonal array design with five input parameters (chitosan type and dosage, pH value, rapid and slow mixing rates) have showed that pH value mainly determined flocculation efficiency, followed by the chitosan type. The flocculation efficiency exhibited a small intensification when increasing the flocculant dosage, whereas rapid mixing speed had a little effect on it. On the other hand, the influence of slow mixing speed is higher compared to the rapid mixing speed. A good flocculation efficiency and high antimicrobial activity indicating synergism between the *Bacillus* sp. BioSol021 biomass and chitosan confirm dual nature of chitosan as a flocculant, as well as a biocontrol agent for potential agricultural applications.

## Figures and Tables

**Figure 1 polymers-14-03282-f001:**
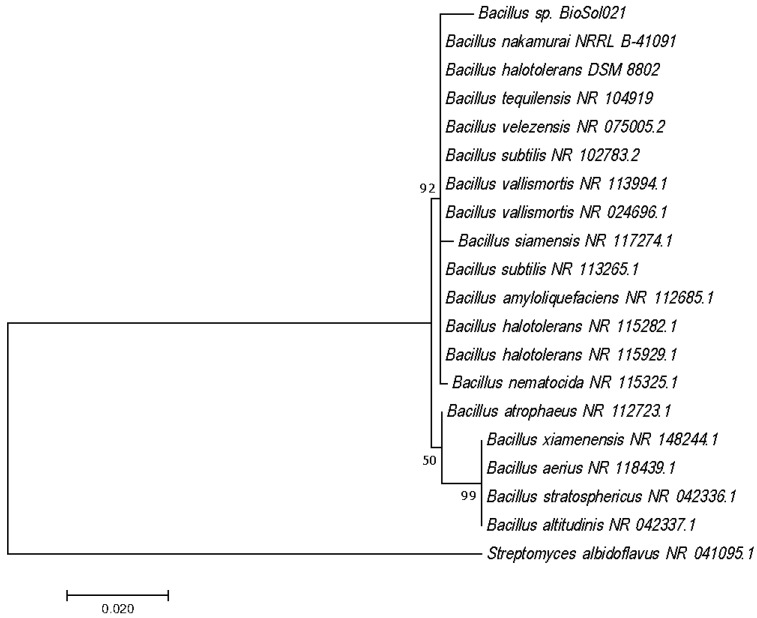
Molecular phylogenetic analysis of the *Bacillus* sp. strain BioSol021 16S rRNA gene sequence by the maximum likelihood method.

**Figure 2 polymers-14-03282-f002:**
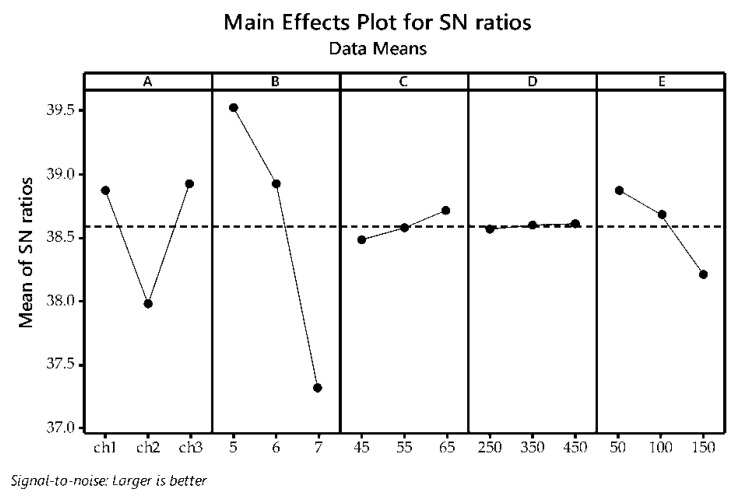
The main effects plot for the S/N analysis of the flocculation efficiency during the flocculation of the *Bacillus* sp. BioSol021 cultivation broth: A—chitosan type, B—pH value, C—chitosan dosage (mg/mL), D—rapid mixing speed (rpm), E—slow mixing speed (rpm).

**Figure 3 polymers-14-03282-f003:**
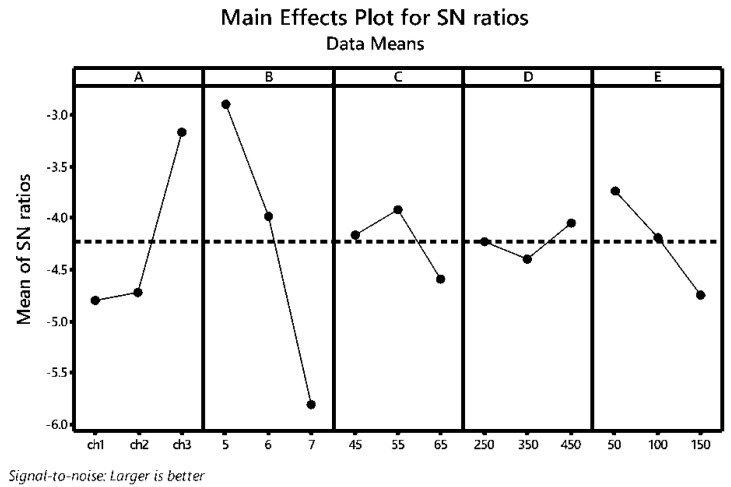
Main effects plot for the S/N analysis of the grey relational grade (GRG) during the flocculation of the *Bacillus* sp. BioSol021 cultivation broth and antimicrobial activity testing against the selected phytopathogens: A—chitosan type, B—pH value, C—chitosan dosage (mg/mL), D—rapid mixing speed (rpm), E—slow mixing speed (rpm).

**Table 1 polymers-14-03282-t001:** The experimental factors and their levels during the flocculation of the *Bacillus* sp. BioSol021 cultivation broth.

Parameter	Notation	Levels		
		1	2	3
Chitosan type	A	ch1	ch 2	ch 3
pH value	B	5	6	7
Chitosan dosage [mg/mL]	C	45	55	65
Rapid mixing [rpm]	D	250	350	450
Slow mixing [rpm]	E	50	100	150

Ch-chitosan; ch 1: ≥75% deacetylated; 100,000–300,000 Da; ch 2: ≥75% deacetylated, 310,000–375,000 Da and ch 3: ≥90% deacetylated, 600,000–800,000 Da.

**Table 2 polymers-14-03282-t002:** The results of the VITEK2 biochemical tests for the producing microorganism *Bacillus* sp. BioSol021.

Parameter	Value	Parameter	Value	Parameter	Value
β-xylosidase	+	Myo-inositol	+	Pyruvate	+
L-Lysine arylamidase	-	Methyl-α-D- glucopyranoside acidification	+	α-glucosidase	+
L-Aspartate arylamidase	-	Ellman	+	D-tagatose	-
Leucine arylamidase	+	Methyl-D-xyloside	-	D-trehalose	+
Phenylalanine arylamidase	+	α-mannosidase	-	Inulin	-
L-Proline arylamidase	-	Maltotriose	-	D-glucose	+
β-galactosidase	-	Glycine arylamidase	+	D-ribose	+
L-Pyrrolidonyl arylamidase	+	D-mannitol	+	Putrescine assimilation	-
α-galactosidase	+	D-mannose	+	Growth in 6.5% NaCl	+
Alanine arylamidase	+	D-melezitose	-	Kanamycin resistance	-
Tyrosine arylamidase	+	N-acetyl-D-glucosamine	-	Oleandomycin resistance	-
β-N-acetyl- glucosaminidase	-	Palatinose	+	Esculin hydrolysis	+
Ala-Phe-Pro arylamidase	-	L-rhamnose	-	Tetrazolium red	+
Cyclodextrin	-	β-glucosidase	+	Polymyxin B resistance	+
D-galactose	-	β-mannosidase	-		
Glycogen	-	Phosphoryl choline	-		

**Table 3 polymers-14-03282-t003:** Composition of the *Bacillus* sp. BioSol021s cultivation broth at the end of the cultivation and at the beginning of the flocculation.

Parameter	Value
Cellulose content (g/L)	3.4
Total nitrogen content (g/L)	0.37
Biomass content (CFU/mL)	2∙10^8^
pH value	7.51

**Table 4 polymers-14-03282-t004:** Flocculation efficiency results using the Taguchi L_18_ orthogonal array during the flocculation of the *Bacillus* sp. BioSol021 cultivation broth.

Run	Parameter Combination	EF (%)	S/N
**1**	**A_1_B_1_C_1_D_1_E_1_**	**98.60**	**39.8776**
2	A_1_B_2_C_2_D_2_E_2_	97.63	39.7918
3	A_1_B_3_C_3_D_3_E_3_	76.63	37.6883
4	A_2_B_1_C_1_D_2_E_2_	89.23	39.0104
5	A_2_B_2_C_2_D_3_E_3_	76.06	37.6230
6	A_2_B_3_C_3_D_1_E_1_	73.76	37.3566
7	A_3_B_1_C_2_D_1_E_3_	97.27	39.7598
8	A_3_B_2_C_3_D_2_E_1_	97.60	39.7886
9	A_3_B_3_C_1_D_3_E_2_	79.83	38.0431
10	A_1_B_1_C_3_D_3_E_2_	95.55	39.6045
11	A_1_B_2_C_1_D_1_E_3_	86.36	38.7263
12	A_1_B_3_C_2_D_2_E_1_	75.27	37.5323
13	A_2_B_1_C_2_D_3_E_1_	94.19	39.4797
14	A_2_B_2_C_3_D_1_E_2_	83.35	38.4176
15	A_2_B_3_C_1_D_2_E_3_	63.10	36.0008
16	A_3_B_1_C_3_D_2_E_3_	93.90	39.4531
17	A_3_B_2_C_1_D_3_E_1_	91.56	39.2346
18	A_3_B_3_C_2_D_1_E_2_	72.97	37.2631

A—chitosan type, B—pH value, C—chitosan dosage (mg/mL), D—rapid mixing speed (rpm), E—slow mixing speed (rpm), EF—flocculation efficiency (%), S/N—signal-to-noise ratio.

**Table 5 polymers-14-03282-t005:** Analysis of variance for the S/N ratios of flocculation efficiency during the flocculation of the *Bacillus* sp. BioSol021 cultivation broth.

Source	Degree of Freedom	Sum of Squares	Mean Square	F-Value	*p*-Value
**A**	**2**	**3.3618**	**1.68088**	**7.72**	**0.017**
**B**	**2**	**15.7743**	**7.88714**	**36.24**	**0.000**
C	2	0.1697	0.08483	0.39	0.691
D	2	0.0064	0.00318	0.01	0.986
E	2	1.4295	0.71477	3.28	0.099
Error	7	1.5234	0.21763		
Total	17	22.2650			

A—chitosan type, B—pH value, C—chitosan dosage (mg/mL), D—rapid mixing speed (rpm), E—slow mixing speed (rpm).

**Table 6 polymers-14-03282-t006:** Flocculation efficiency and precipitate antimicrobial activity results against the selected bacterial and fungal phytopathogens (L_18_ array) during the flocculation of the *Bacillus* sp. BioSol021 cultivation broth.

Run	EF (%)	Antimicrobial Activity/Inhibition Zone Diameter (mm)			
		*Xanthomonas* sp.	*Colletotrichum* sp.	*Fusarium* sp.	*Aspergillus* sp.
	Y1	Y2	Y3	Y4	Y5
1	98.60	42.00	40.33	36.33	36.33
2	97.63	42.67	34.67	37.00	32.33
3	76.63	22.33	29.33	35.33	41.00
4	89.23	43.33	35.00	35.00	41.67
5	76.06	43.00	35.00	35.00	39.33
6	73.76	41.33	33.00	34.33	35.33
7	97.27	40.33	41.00	36.67	44.33
8	97.60	40.00	38.33	37.67	41.33
9	79.83	43.67	39.33	34.67	40.67
10	95.55	29.00	32.00	38.00	26.00
11	86.36	31.00	32.67	36.67	33.00
12	75.27	32.33	32.67	35.00	29.67
13	94.19	40.00	41.00	37.00	37.33
14	83.35	37.33	34.67	35.67	35.33
15	63.10	31.33	32.00	31.33	40.67
16	93.90	40.00	35.00	35.33	40.33
17	91.56	44.67	35.33	34.67	40.00
18	72.97	34.00	31.67	34.00	43.67

Y1—flocculation efficiency (%), Y2 to Y5—inhibition zone diameters (mm) against the selected phytopathogens.

**Table 7 polymers-14-03282-t007:** Original experimental data after normalization and their corresponding deviations during the flocculation of the *Bacillus* sp. BioSol021 cultivation broth and antimicrobial activity testing against the selected phytopathogens.

Run	Normalized Values					Deviation Sequences				
	Y1	Y2	Y3	Y4	Y5	Y1	Y2	Y3	Y4	Y5
1	1.0000	0.8806	0.9429	0.7500	0.5636	0.0000	0.1194	0.0571	0.2500	0.4364
2	0.9727	0.9104	0.4571	0.8500	0.3455	0.0273	0.0896	0.5429	0.1500	0.6545
3	0.3812	0.0000	0.0000	0.6000	0.8182	0.6188	1.0000	1.0000	0.4000	0.1818
4	0.7361	0.9403	0.4857	0.5500	0.8545	0.2639	0.0597	0.5143	0.4500	0.1455
5	0.3650	0.9254	0.4857	0.5500	0.7273	0.6350	0.0746	0.5143	0.4500	0.2727
6	0.3003	0.8507	0.3143	0.4500	0.5091	0.6997	0.1493	0.6857	0.5500	0.4909
7	0.9626	0.8060	1.0000	0.8000	1.0000	0.0374	0.1940	0.0000	0.2000	0.0000
8	0.9717	0.7910	0.7714	0.9500	0.8364	0.0283	0.2090	0.2286	0.0500	0.1636
9	0.4712	0.9552	0.8571	0.5000	0.8000	0.5288	0.0448	0.1429	0.5000	0.2000
10	0.9141	0.2985	0.2286	1.0000	0.0000	0.0859	0.7015	0.7714	0.0000	1.0000
11	0.6552	0.3881	0.2857	0.8000	0.3818	0.3448	0.6119	0.7143	0.2000	0.6182
12	0.3428	0.4478	0.2857	0.5500	0.2000	0.6572	0.5522	0.7143	0.4500	0.8000
13	0.8756	0.7910	1.0000	0.8500	0.6182	0.1244	0.2090	0.0000	0.1500	0.3818
14	0.5703	0.6716	0.4571	0.6500	0.5091	0.4297	0.3284	0.5429	0.3500	0.4909
15	0.0000	0.4030	0.2286	0.0000	0.8000	1.0000	0.5970	0.7714	1.0000	0.2000
16	0.8675	0.7910	0.4857	0.6000	0.7818	0.1325	0.2090	0.5143	0.4000	0.2182
17	0.8018	1.0000	0.5143	0.5000	0.7636	0.1982	0.0000	0.4857	0.5000	0.2364
18	0.2781	0.5224	0.2000	0.4000	0.9636	0.7219	0.4776	0.8000	0.6000	0.0364

Y1-flocculation efficiency (%), Y2 to Y5-inhibition zone diameters (mm) against the selected phytopathogens: *Xanthomonas* sp., *Colletotrichum* sp., *Fusarium* sp. and *Aspegillus* sp., respectively.

**Table 8 polymers-14-03282-t008:** Grey relational coefficient, grey relational grade (GRG) and S/N conversion of GRG for different operational parameters combinations during the flocculation of the *Bacillus* sp. BioSol021 cultivation broth and antimicrobial activity testing against the selected phytopathogens.

Run	Grey Relational Coefficient					GRG	S/N	Rank
	Y1	Y2	Y3	Y4	Y5			
1	1.0000	0.8072	0.8974	0.6667	0.5340	0.7811	−2.14629	3
2	0.9482	0.8481	0.4795	0.7692	0.4331	0.6956	−3.15261	5
3	0.4469	0.3333	0.3333	0.5556	0.7333	0.4805	−6.36628	16
4	0.6545	0.8933	0.4930	0.5263	0.7746	0.6684	−3.49982	8
5	0.4405	0.8701	0.4930	0.5263	0.6471	0.5954	−4.50383	11
6	0.4168	0.7701	0.4217	0.4762	0.5046	0.5179	−5.71558	15
**7**	**0.9304**	**0.7204**	**1.0000**	**0.7143**	**1.0000**	**0.8730**	**−1.17951**	**1**
8	0.9464	0.7053	0.6863	0.9091	0.7534	0.8001	−1.93719	2
9	0.4860	0.9178	0.7778	0.5000	0.7143	0.6792	−3.36039	7
10	0.8533	0.4161	0.3933	1.0000	0.3333	0.5992	−4.44838	10
11	0.5919	0.4497	0.4118	0.7143	0.4472	0.5229	−5.63086	14
12	0.4321	0.4752	0.4118	0.5263	0.3846	0.4460	−7.01354	18
13	0.8008	0.7053	1.0000	0.7692	0.5670	0.7685	−2.28754	4
14	0.5378	0.6036	0.4795	0.5882	0.5046	0.5427	−5.30826	12
15	0.3333	0.4558	0.3933	0.3333	0.7143	0.4460	−7.01333	17
16	0.7906	0.7053	0.4930	0.5556	0.6962	0.6481	−3.76703	9
17	0.7161	1.0000	0.5072	0.5000	0.6790	0.6805	−3.34368	6
18	0.4092	0.5115	0.3846	0.4545	0.9322	0.5384	−5.37790	13

Y1—flocculation efficiency (%), Y2 to Y5—inhibition zone diameters (mm) against the selected phytopathogens: *Xanthomonas* sp., *Colletotrichum* sp., *Fusarium* sp. and *Aspegillus* sp., respectively.

**Table 9 polymers-14-03282-t009:** Analysis of variance for S/N ratios of grey relational grade (GRG) during the flocculation of the *Bacillus* sp. BioSol021 cultivation broth and antimicrobial activity testing against the selected phytopathogens.

Scheme.	Degree of Freedom	Sum of Squares	Mean Square	F-Value	*p*-Value
A	2	10.2073	5.1037	3.01	0.114
**B**	**2**	**26.1180**	**13.0590**	**7.71**	**0.017**
C	2	1.3837	0.6918	0.41	0.680
D	2	0.3583	0.1791	0.11	0.901
E	2	3.0274	1.5137	0.89	0.451
Error	7	11.8635	1.6948		
Total	17	52.9581			

A—chitosan type, B—pH value, C—chitosan dosage (mg/mL), D—rapid mixing speed (rpm), E—slow mixing speed (rpm).

**Table 10 polymers-14-03282-t010:** Comparison of the optimization and experimental validation results during the flocculation of the *Bacillus* sp. BioSol021 cultivation broth and antimicrobial activity testing against the selected phytopathogens.

Output	Initial Controllable Parameters	Optimal Controllable Parameters	
		Prediction	Experiment
Parameter Set	A_3_B_1_C_2_D_1_E_3_	A_3_B_1_C_2_D_3_E_1_	A_3_B_1_C_2_D_3_E_1_
Y1	97.27	–	97.50
Y2	40.33	–	43.00
Y3	41.00	–	40.33
Y4	36.67	–	37.00
Y5	44.33		44.85
GRG	0.8730	0.8673	0.8956

A—chitosan type, B—pH value, C—chitosan dosage (mg/mL), D—rapid mixing speed (rpm), E—slow mixing speed (rpm); Y1—flocculation efficiency (%), Y2 to Y5—inhibition zone diameters (mm) against the selected phytopathogens: *Xanthomonas* sp., *Colletotrichum* sp., *Fusarium* sp. and *Aspegillus* sp., respectively; GRG—grey relational grade.

## Data Availability

Not applicable.

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
