# Peer review of "Taguchi Grey Relational Analysis for Multi-Response Optimization of Bacillus Bacteria Flocculation Recovery from Fermented Broth by Chitosan to Enhance Biocontrol Efficiency"

_polymers, 2022, doi:10.3390/polym14163282_

Round 1
Reviewer 1 Report
This paper dealt with Taguchi Grey Relational Analysis for Multi-Response Optimization of Bacillus bacteria flocculation recovery from fermented broth by chitosan to enhance biocontrol efficiency. The following questions need to be clarified.
L17-19: ~Experiments were performed based on L18 standard Taguchi orthogonal array design with five input parameters (chitosan type and dosage, pH value and mixing rate).~ Q: There are only four input parameters (chitosan type and dosage, pH value and mixing rate).
Tables 1, 4, 5, 6, 7 and 8: Q: The tables must be intelligible without reference to the text, so be sure to include an explanation of every abbreviation.
Author Response
Reviewer 1
Comments and Suggestions for Authors
This paper dealt with Taguchi Grey Relational Analysis for Multi-Response Optimization of Bacillus bacteria flocculation recovery from fermented broth by chitosan to enhance biocontrol efficiency. The following questions need to be clarified.
Q1
L17-19: ~Experiments were performed based on L18 standard Taguchi orthogonal array design with five input parameters (chitosan type and dosage, pH value and mixing rate).
~ Q: There are only four input parameters (chitosan type and dosage, pH value and mixing rate).
A1
There were two mixing rates values, slow and rapid (levels are given in Table 1). In the abstract the mentioned lines were corrected according to suggestion to clearly indicate the number of the input variables.
Experiments were performed based on L18 standard Taguchi orthogonal array design with five input parameters (chitosan type and dosage, pH value, rapid and slow mixing rates).
Q2
Tables 1, 4, 5, 6, 7 and 8: Q: The tables must be intelligible without reference to the text, so be sure to include an explanation of every abbreviation.
A2
An explanation of every abbreviation given in the aforementioned Tables is added.
Reviewer 2 Report
The manuscript "Taguchi Grey Relational Analysis for Multi-Response Optimization of Bacillus bacteria flocculation recovery from fermented broth by chitosan to enhance biocontrol efficiency" presents a study aimed to investigate the dual nature of chitosan as antimicrobial and flocculation agent. This study was performed using a L18 standard Taguchi orthogonal array design with five input parameters (chitosan type and dosage, pH value and mixing rate). The manuscript is well written and it is very interesting. The introduction is exhaustive and clear. It might be reduced delating some repetition of the same concepts but it is not essential. Also the experimental method and results are clearly described and supported by literature. The conclusions arr consistent with the obtained results. In my opinion the manuscript can be pubblished in the present form or with minor revision.
Author Response
Reviewer 2
The manuscript "Taguchi Grey Relational Analysis for Multi-Response Optimization of Bacillus bacteria flocculation recovery from fermented broth by chitosan to enhance biocontrol efficiency" presents a study aimed to investigate the dual nature of chitosan as antimicrobial and flocculation agent. This study was performed using a L18 standard Taguchi orthogonal array design with five input parameters (chitosan type and dosage, pH value and mixing rate). The manuscript is well written and it is very interesting. The introduction is exhaustive and clear. It might be reduced delating some repetition of the same concepts but it is not essential. Also the experimental method and results are clearly described and supported by literature. The conclusions are consistent with the obtained results. In my opinion the manuscript can be pubblished in the present form or with minor revision.
The authors are glad to hear the positive comments regarding the manuscript quality from the Reviewer.
Reviewer 3 Report
The authors presented an exciting problem optimization of parameter values influencing the flocculation efficiency and antimicrobial activity of bacterial aggregates of the genus Bacillus and chitosan. For this purpose, they used Taguchi Grey Relational Analysis for Multi-Response Optimization.
The experiment was well planned and fully implemented.
I would suggest specifying a computer program used to analyze the obtained results in the methodology.
Author Response
Reviewer 3
The authors presented an exciting problem optimization of parameter values influencing the flocculation efficiency and antimicrobial activity of bacterial aggregates of the genus Bacillus and chitosan. For this purpose, they used Taguchi Grey Relational Analysis for Multi-Response Optimization.
The experiment was well planned and fully implemented.
I would suggest specifying a computer program used to analyze the obtained results in the methodology.
Thank you for the suggestion. It was an error not to include information about the software used. The following line is added to the manuscript text:
The Minitab software package (version 17.0) was used for statistical analyses.